# Association between Dietary Diversity Score and Metabolic Syndrome in Korean Adults: A Community-Based Prospective Cohort Study

**DOI:** 10.3390/nu14245298

**Published:** 2022-12-13

**Authors:** Jiyeon Kim, Minji Kim, Yoonjin Shin, Jung-Hee Cho, Donglim Lee, Yangha Kim

**Affiliations:** 1Department of Nutritional Science and Food Management, Ewha Womans University, Seoul 03760, Republic of Korea; 2Graduate Program in System Health Science and Engineering, Ewha Womans University, Seoul 03760, Republic of Korea; 3Fisheries Policy Research Division, Korea Maritime Institute, Busan 49111, Republic of Korea

**Keywords:** dietary diversity score, dietary quality, metabolic syndrome, longitudinal study, gender stratification

## Abstract

Dietary diversity is recognized as a key indicator of dietary quality and is known to affect the burden of non-communicable diseases. This study examined the gender-stratified association between dietary diversity score (DDS) and risk of metabolic syndrome (MetS) in 5468 adults aged 40–69 years during a 12-year follow-up of the Korean Genome and Epidemiology Study (KoGES). DDS was calculated according to the consumption of the five food groups based on the Dietary Reference Intakes (DRIs) for Koreans. The Cox proportional hazard model was used to evaluate MetS risk according to DDS. A higher DDS was negatively associated with the consumption of grains but positively associated with the consumption of fruits and non-salted vegetables. Furthermore, participants with a higher DDS showed higher consumption of fish and milk. Prospectively, a higher DDS was significantly associated with a lower risk of MetS in men (HR: 0.76, 95% CI: 0.63–0.92, *p* < 0.01). In all participants, a higher DDS was inversely associated with the incidence of abdominal obesity (men, HR: 0.76, 95% CI: 0.62–0.93, *p* < 0.01; women, HR: 0.79, 95% CI: 0.67–0.94, *p* < 0.01). Furthermore, men with a higher DDS had a lower risk of hypertriglyceridemia (HR: 0.83, 95% CI: 0.71–0.97, *p* < 0.05). These findings suggested that eating a more varied diet might have favorable effects on preventing MetS in Korean adults.

## 1. Introduction

Metabolic syndrome (MetS) is a collection of various conditions, including abdominal obesity, hypertension, impaired fasting glycemia, and dyslipidemia (reduced high-density lipoprotein cholesterol [HDL-C] and increased triglycerides [TG]), that are major risk factors for cardiovascular disease and type 2 diabetes [1,2]. The prevalence of MetS is rapidly rising worldwide. According to the International Diabetes Federation (IDF) criteria, the prevalence of MetS in the US was 33–39% from 2003 to 2012 [3]. The 2021 Metabolic Syndrome Fact Sheet, based on the Korean National Health and Nutrition Examination Surveys (KNHANES) data, reported that the prevalence of MetS increased from 21.6% in 2007 to 22.9% in 2018 [4]. The modifiable lifestyle factors associated with MetS include sedentary behavior, smoking, alcohol consumption, and diet [5].

The dietary diversity score (DDS) is an indicator for assessing diet by counting the number of major food groups consumed, reflecting overall dietary quality compared with the evaluation of single foods or nutrients [6,7]. Epidemiological studies have indicated that a higher DDS is intimately related to a healthier diet with adequate intake of all food groups, which may provide a wide range of macro- and micronutrients [8,9]. Furthermore, the consumption of a more diverse diet has shown protective effects against non-communicable diseases [10,11].

Gender differences have been reported for dietary intake. A systematic review suggested gender differences in food choice and eating behavior and emphasized the importance of considering a gender-specific approach when addressing nutrition issues in research [12]. More specifically, women tend to consume more fruits and vegetables and show higher levels of dietary restraint compared to men [13,14].

According to a 6-year follow-up in the Korean population of middle-aged and older adults, factor scores for the healthy dietary pattern, characterized by a variety of healthy food choices, were inversely associated with MetS risk [15]. Similarly, a more diversified diet was found to be associated with a lower risk of MetS and some MetS components in the US population, based on the US Healthy Food Diversity Index [16]. However, a meta-analysis reported no significant association between DDS and most of the cardiometabolic risk factors, such as obesity, lipid profile, blood pressure (BP), diabetes, and MetS [6].

Therefore, we aimed to investigate the association between DDS and the risk of developing MetS and its components in middle-aged and older adults stratified by gender, using data from the Korean Genome and Epidemiology Study (KoGES), a large community-based cohort study.

## 2. Materials and Methods

### 2.1. Study Population

We used data from a prospective community-based Ansan–Ansung cohort study, which is part of the KoGES, to investigate dietary, environmental, and lifestyle factors and chronic diseases among Koreans [17]. Detailed information on this study design and procedures of the KoGES was described in a previous report. In brief, 10,030 Korean adults aged 40–69 years were recruited from the Ansan (urban) and Ansung (rural) areas at baseline between 2001 and 2002. Follow-up examinations were conducted biennially. This study was based on the data from the baseline through the sixth follow-up examination between 2013 and 2014. Of the original 10,030 participants, we excluded participants with a history of MetS, diabetes, hypertension, and dyslipidemia (*n* = 4100) and those who reported implausible total daily energy intake (<500 kcal/day or >5000 kcal/day, *n* = 229). An additional 233 participants with missing information on anthropometric, demographic, lifestyle, biochemical, and menopausal status were excluded. Finally, 5468 participants (2824 men and 2644 women) were analyzed. The study was approved by the Institutional Review Board of Ewha Womans University (2021-0316, October 2021).

### 2.2. DDS

To assess dietary diversity, we obtained data on food consumption using the semi-quantitative food frequency questionnaire (FFQ) developed for the KoGES [18]. This FFQ included 103 food items. Food consumption was measured once, at the baseline of this study, concerning the participant’s dietary intake over the past year. Participants were asked to report their average food frequency (on a 9-point scale of “almost none”, “once a month”, “twice or three times a month”, “once or twice a week”, “twice or three times a week”, “five or six times a week”, “once a day”, “twice a day”, and “three times a day”) and the average portion size (on a 3-point scale of “0.5 times the reference”, “reference”, and “1.5–2.0 times the reference”) for each food item for 1 year. The duration of seasonal variety of fruit intake was divided into four categories (3, 6, 9, and 12 months). The validity and reproducibility of the FFQ have been described elsewhere [18].

In this study, we measured the DDS modified by Conklin et al. [19], based on five major food groups in the dietary guideline of Dietary Reference Intakes (DRIs) for Koreans: (1) grains; (2) meat, fish, eggs, and beans; (3) vegetables; (4) fruits; and (5) milk [20]. The five major food groups were further subdivided into 23 food groups for South Korea (Appendix A), based on the dietary quality questionnaire that was designed for use in rapid assessment without requiring nutrition expertise to administer [21,22]. The DDS was calculated by counting the food items of five major food groups consumed at least once per week. Specifically, the food items consumed were counted as 1 point except for “almost none”, “once a month”, and “twice or three times a month”, based on the reported frequency from the FFQ. Each food group was only counted once, regardless of the number of food items from the same group, when calculating DDS. Each time another food group was consumed, the DDS increased by 1 point. To examine the effect of food groups by subtype, grains were subclassified into whole grains and refined grains. Meat, fish, eggs, and beans were subclassified into meat, fish, and others. In addition, vegetables were subclassified into non-salted vegetables and salted vegetables. The total DDS ranges from 0 to 5. A higher DDS reflects a more diverse diet. Participants were divided into 3 groups according to DDS: ≤3, 4, and 5.

### 2.3. Measurements

Anthropometric measurements were performed by trained research staff at each follow-up visit. Height and body weight were measured with the participants wearing a patient gown and barefoot. Body mass index (BMI) was calculated as body weight (kg) divided by the square of height (m^2^). BP was measured in both arms using a mercury sphygmomanometer (W.A. Baum Co., Inc., Copiague, NY, USA) after resting for at least 5 min. This study used the mean value of repeated measurements to define systolic blood pressure (SBP) and diastolic blood pressure (DBP). Blood samples were obtained after at least 8 h of fasting, and plasma was separated for biochemical measurements. The plasma concentrations of glucose, TGs, and HDL-C were enzymatically measured using an autoanalyzer (Bayer HealthCare, Berlin, Germany).

### 2.4. Definition of MetS

We used the diagnostic criteria for the MetS based on the Harmonized Joint Scientific Statement [23]. Thus, MetS was defined as the presence of three or more of the following: (1) abdominal obesity (waist circumference ≥ 90 cm in men or ≥80 cm in women); (2) elevated BP (≥130/85 mmHg, physician’s diagnosis of hypertension or medication for elevated BP); (3) hypertriglyceridemia (serum TG concentration ≥ 150 mg/dL, medication for elevated TG); (4) elevated fasting glucose (fasting blood glucose ≥ 100 mg/dL, physician’s diagnosis of type 2 diabetes, medication for elevated glucose or use of insulin); and (5) reduced HDL-C (serum HDL-C concentration < 40 mg/dL in men or <50 mg/dL in women, medication for reduced HDL-C).

### 2.5. Covariates

The demographic characteristics, socioeconomic status, and lifestyle factors of the participants were surveyed at baseline. Covariates included age, regular exercise (<30 min/day, ≥30 min/day), current alcohol consumption (yes, no), current smoking status (yes, no), education level (<high school, ≥high school), monthly household income (<2 million KRW, ≥2 million KRW), menopausal status (pre-, post-), family history of diabetes, hypertension, and dyslipidemia (yes, no for each), and BMI.

### 2.6. Statistical Analysis

Continuous variables are expressed as mean and standard error (SE), and categorical variables are expressed as numbers and percentages. The generalized linear model (GLM) and the Chi-square test were applied to determine the differences in means and distribution of general characteristics and to test the linear trends according to DDS. The multivariable Cox proportional hazard model was used to estimate the hazard ratios (HRs) and 95% confidence intervals (CIs) for the risk of MetS and its components according to DDS during the follow-up. Model 1 was adjusted for age, BMI, and energy intake, and Model 2 was adjusted for all variables in Model 1 plus education level and household income. Model 3 was adjusted for all variables in Model 2 plus regular exercise, alcohol consumption, smoking status, and family history of diabetes, hypertension, and dyslipidemia. Women’s data were additionally adjusted for menopausal status. When selecting variables for adjustment in the multivariable model, potential confounders from the previously published scientific literature were taken into account [5] with stepwise regression procedures. All analyses were performed using SAS software, version 9.4 (SAS Institute, Cary, NC, USA). Statistical significance was considered at *p* < 0.05. We stratified the analysis by gender, as previous research reported gender differences in dietary intake and the prevalence of MetS [24,25].

## 3. Results

### 3.1. Baseline Characteristics

Table 1 describes the characteristics of participants according to DDS at baseline. In both men and women, participants with higher DDS were younger, less physically active, more educated, and had a higher household income (all *p* < 0.01). Men with a higher DDS were less likely to be current smokers (*p* < 0.01). Smoking status was not significantly associated with DDS among women. Additionally, men with a higher DDS had a higher BMI (*p* < 0.05). Women did not show any difference in BMI among groups. Men and women with a higher DDS had lower SBP and DBP (all *p* < 0.05). Regarding biochemical profiles, men with a higher DDS had a higher concentration of low-density lipoprotein cholesterol (LDL-C) (*p* < 0.01) and a lower concentration of TG (*p* < 0.05). However, women did not show any difference in biochemical profiles among groups.

### 3.2. Associations between DDS and Food Consumption

Food consumption according to DDS is shown in Table 2. In all participants, a higher DDS was associated with higher consumption of meat, fish, eggs, and beans (*p* < 0.01), fruits (*p* < 0.01), and milk (*p* < 0.01). By contrast, a higher DDS was associated with lower consumption of grains (*p* < 0.01) despite higher consumption of other food groups. Consumption of total vegetables was not significantly associated with DDS. When vegetables were classified into subgroups of salted vegetables and non-salted vegetables, however, a higher DDS was associated with lower consumption of salted vegetables (*p* < 0.01) but higher consumption of non-salted vegetables (*p* < 0.01).

### 3.3. Associations between DDS and Nutrient Intake

The associations of DDS with nutrient intakes per 1000 kcal are presented in Table 3. In both men and women, participants with a higher DDS showed a higher total energy intake (*p* < 0.01). Men and women with a higher DDS showed a lower carbohydrate intake despite a higher total energy intake (*p* < 0.01). In both men and women, there was a positive association between the DDS and most of the nutrient intakes (all *p* < 0.01). However, sodium intake did not differ significantly with DDS among men and women.

### 3.4. Longitudinal Association of DDS with the Risk of MetS and Its Components

The risks for MetS and its components, according to DDS, are shown in Table 4. In men, a higher DDS was associated with a 24% lower risk of MetS (HR: 0.76, 95% CI: 0.63–0.92, *p* < 0.01) after adjusting for age, energy intake, regular exercise, education level, household income, alcohol consumption, smoking status, family history of diabetes, hypertension, and dyslipidemia, and BMI. Men with a higher DDS also had a lower risk of abdominal obesity (HR: 0.76, 95% CI: 0.62–0.93, *p* < 0.01) and hypertriglyceridemia (HR: 0.83, 95% CI: 0.71–0.97, *p* < 0.05) after adjusting for all confounding factors. There was no association between DDS and the risk of incident MetS in women. In women, a higher DDS was associated with a lower risk of abdominal obesity (HR: 0.79, 95% CI: 0.67–0.94, *p* < 0.01) only after adjusting for all confounding factors. Changes in body weight and metabolic syndrome components during follow-up are summarized in Appendix A.

## 4. Discussion

In this prospective cohort study, we found that a higher DDS was associated with a lower risk of incident MetS and its components, such as abdominal obesity and hypertriglyceridemia, after adjustment for potential confounders in men. A higher DDS was associated with a lower risk of abdominal obesity in both men and women. To the best of our knowledge, this study is the first to reveal the associations between DDS and MetS and its components in Korean adults.

In our study, a higher DDS was longitudinally associated with a decreased risk of MetS in men only, not women. According to the Seguimiento Universidad de Navarra (SUN) prospective cohort, higher dietary scores, which reflect the Mediterranean dietary pattern, were inversely associated with MetS incidence during a 6-year follow-up period [26]. In a cross-sectional study among Tehranian adults aged > 18 years, DDS had an inverse association with MetS and some of its components after adjustment for confounding factors such as age, total energy intake, energy from fat, BP with use of estrogen medication, smoking, physical activity, and BMI [10]. Moreover, a cross-sectional study on DDS and MetS features in adults with prediabetes found that a higher DDS was associated with a low probability of MetS [27]. We found that DDS was inversely associated with the risk of individual components of MetS as well. A cross-sectional study conducted among Iranian female students suggested that DDS is associated with a lower risk of abdominal adiposity [28]. Another cross-sectional study showed that individuals in the higher DDS quartiles had lower serum TG concentrations than those in the lower quartiles [29]. In addition, a case–control study found that a higher DDS was associated with a lower level of TG and a smaller waist circumference [27].

DDS is a useful indicator for assessing overall diet quality, nutrient adequacy, and diet–disease associations [30]. A higher DDS was associated with a healthier diet, with those in the upper category consuming less refined grains [28]. Similarly, our data showed that a higher DDS was characterized by lower consumption of grains, especially refined grains, in both men and women. Several potential mechanisms have been proposed to explain the inverse association between DDS and the risk of MetS and its components. Foods that are quickly digested in the gut, including most refined-grain products, have a relatively high glycemic index (GI), whereas whole-grain products, non-starchy vegetables, legumes, and fruits tend to have a low GI [31,32]. A high-GI diet may affect appetite and nutrient partitioning in a way that promotes body fat storage, resulting in excess weight gain [33]. A cross-sectional study among Iranian adults linked a higher dietary GI with an increased risk of abdominal obesity [34]. A previous study reported that neither obesity nor insulin resistance, which are mechanisms underlying metabolic disease, are caused by dietary fat intake but by excessive intake of carbohydrates in Asian populations. In Korean male adults aged 30–65 years, the percentage of energy from carbohydrates was positively associated with the prevalence of MetS and its components [35]. In addition, high consumption of refined grains was significantly associated with elevated TG and fasting glucose levels in an Asian-Indian population [36]. It can be suggested that those with a higher DDS ate less refined grains and carbohydrates, partly contributing to a lower risk of MetS and its components after 12 years.

Moreover, we found that all participants with a higher DDS had a higher consumption of fruits. Fruits contain many health-promoting components, including antioxidant nutrients, phytochemicals, and dietary fiber [37]. In general, individuals with MetS exhibited elevated levels of oxidative damage as a result of the overproduction of reactive oxygen species (ROS) [38]. Various nutritional antioxidants from fruits, such as vitamin C, vitamin E, and β-carotene, might reduce the risk of MetS by decreasing ROS production and neutralizing free radicals [39]. The intakes of total vitamin A and total vitamin C, as well as moderate and high fruit intakes, were associated with a lower prevalence of MetS in Korean women [38]. In addition, a high intake of dietary fiber, commonly obtained through consumption of fruits and vegetables, was reported to suppress oxidative stress and inflammation, both of which have been linked to the development of MetS [40,41]. A meta-analysis showed that dietary fiber intake is associated with a lower likelihood of MetS [42].

Vegetables are rich sources of antioxidants, such as carotenoids, vitamin C, folate, and selenium, as well as fiber and phytochemicals that play an important role in the etiology of diseases [43]. Our study showed that the consumption of vegetables was not significantly associated with DDS. When vegetables were classified into subgroups of salted vegetables and non-salted vegetables, however, a higher DDS was associated with lower consumption of salted vegetables but higher consumption of non-salted vegetables. This finding is similar to the results of another study that found the “white rice and salted vegetables” pattern was inversely associated with the food variety score [44]. In Korean adults aged ≥ 19 years, low intake of non-salted vegetables was positively associated with MetS in the total participants [45]. In our study, participants with a high DDS showed increased fish consumption. The consumption of fish, which provides a major source of marine nutrients such as docosahexaenoic acid and eicosapentaenoic acid (EPA), has been identified as a protective factor against several types of disease [46]. Specifically, favorable relationships between these fatty acids from fish and MetS have been reported [47]. The eicosanoids derived from EPA could be involved in suppressing an inflammatory response, ultimately preventing the onset of MetS [48,49]. In addition, a higher DDS was strongly associated with higher milk consumption. High consumption of dairy products might have preventive effects against chronic disease because they contain abundant proteins, essential amino acids, calcium, phosphorous, potassium, vitamin A, and vitamin D [50,51]. A cross-sectional study conducted among Tehranian adults suggested that dairy consumption was inversely associated with the risk of having MetS [52].

We observed an inverse association between DDS and MetS in men only, not women, after adjusting for socioeconomic factors such as education level and household income. Gender differences were found regarding the association between socioeconomic factors and the prevalence of MetS [53,54]. MetS is characterized by a constellation of cardiac risk factors that include abdominal obesity, atherogenic dyslipidemia, hypertension, and insulin resistance. There are several remarkable features of the MetS in women. Menopause heralds a reduction in circulating levels of estrogen, which possibly affects adiposity, lipid metabolism, and prothrombotic state, consequently contributing to increased cardiovascular risk [55]. These features might relate to differences between men and women in the development of MetS.

This study has several strengths. It is the first to investigate the associations between DDS and MetS in Korean adults in a prospective study with a large sample size and long-term follow-up. However, our study also has some limitations. First, we assessed food consumption only at the baseline and did not determine whether the dietary patterns of participants had changed during the follow-up time. Second, residual confounding, or confounding by unmeasured factors that were not considered in the analysis, might affect the metabolic risk.

## 5. Conclusions

In conclusion, this prospective study showed that a higher DDS was associated with a lower risk of incident MetS and its components (i.e., abdominal obesity, hypertriglyceridemia) in men. Therefore, it can be suggested that eating a more varied diet with food items from each of the five main food groups might help to prevent the development of MetS and its components in middle-aged and older Korean adults.

## Figures and Tables

**Table 1 nutrients-14-05298-t001:** Baseline characteristics of participants according to dietary diversity score.

	Men				Women			
	≤3 (*n* = 524)	4 (*n* = 1156)	5 (*n* = 1144)	*p*-Value	≤3 (*n* = 297)	4 (*n* = 925)	5 (*n* = 1422)	*p*-Value
Age (years)	52.4 ± 0.39	51.4 ± 0.26	49.8 ± 0.24	<0.001	51.7 ± 0.54	50.2 ± 0.28	48.7 ± 0.21	<0.001
Regular exercise (%)	325 (62.0)	694 (60.0)	687 (60.1)	0.001	179 (60.3)	489 (52.9)	779 (54.8)	<0.001
Current alcohol consumption (%)	378 (72.1)	843 (72.9)	815 (71.2)	0.763	73 (24.6)	287 (31.0)	442 (31.1)	0.104
Current smoking status (%)	289 (55.2)	620 (53.6)	562 (49.1)	<0.001	12 (4.04)	43 (4.65)	42 (2.96)	0.139
Education level (≥high school, %)	258 (49.3)	641 (55.5)	738 (64.5)	<0.001	82 (27.6)	324 (35.0)	699 (49.1)	<0.001
Household income (>2 million KRW, %)	180 (34.4)	456 (39.4)	548 (47.9)	<0.001	58 (19.5)	312 (33.7)	645 (45.4)	<0.001
Postmenopausal status (%)	-	-	-	-	165 (55.6)	473 (51.1)	664 (46.7)	0.007
Family history of diabetes, hypertension, and dyslipidemia (%)	109 (20.8)	258 (22.3)	245 (21.4)	0.755	75 (25.2)	244 (26.3)	387 (27.2)	0.756
BMI (kg/m^2^)	23.3 ± 0.12	23.3 ± 0.08	23.5 ± 0.08	0.037	24.0 ± 0.18	23.7 ± 0.10	23.8 ± 0.08	0.474
SBP (mmHg)	119.5 ± 0.65	117.9 ± 0.44	117.0 ± 0.44	0.008	113.9 ± 0.89	113.0 ± 0.50	111.8 ± 0.41	0.043
DBP (mmHg)	80.3 ± 0.44	79.4 ± 0.30	78.9 ± 0.30	0.026	76.0 ± 0.58	74.5 ± 0.33	73.9 ± 0.27	0.003
Total cholesterol (mg/dL)	187.2 ± 1.55	188.4 ± 1.04	191.4 ± 1.05	0.072	184.4 ± 1.96	184.9 ± 1.11	186.3 ± 0.90	0.308
HDL-C (mg/dL)	45.8 ± 0.44	45.6 ± 0.29	45.7 ± 0.30	0.869	48.4 ± 0.58	48.8 ± 0.33	48.9 ± 0.27	0.705
LDL-C (mg/dL)	109.9 ± 1.49	112.6 ± 1.00	116.6 ± 1.01	0.001	113.3 ± 1.74	113.2 ± 0.99	114.9 ± 0.80	0.209
Triglyceride (mg/dL)	157.6 ± 4.18	150.9 ± 2.82	145.8 ± 2.83	0.033	114.0 ± 2.90	114.4 ± 1.64	112.7 ± 1.32	0.777
Fasting glucose (mg/dL)	84.7 ± 0.58	84.9 ± 0.39	85.7 ± 0.39	0.389	82.3 ± 0.60	80.6 ± 0.34	80.8 ± 0.28	0.068
HbA1c (%)	5.57 ± 0.02	5.57 ± 0.01	5.56 ± 0.01	0.989	5.54 ± 0.02	5.49 ± 0.01	5.50 ± 0.01	0.351
hs-CRP (mg/dL)	0.30 ± 0.03	0.23 ± 0.02	0.23 ± 0.02	0.071	0.21 ± 0.02	0.18 ± 0.01	0.18 ± 0.01	0.755
Triglyceride/HDL-C	3.71 ± 0.12	3.59 ± 0.08	3.47 ± 0.08	0.153	2.51 ± 0.09	2.50 ± 0.05	2.47 ± 0.04	0.842
Atherogenic index	3.23 ± 0.05	3.29 ± 0.03	3.35 ± 0.03	0.280	2.91 ± 0.05	2.90 ± 0.03	2.92 ± 0.02	0.776

KRW, Korean won; BMI, body mass index; SBP, systolic blood pressure; DBP, diastolic blood pressure; HDL-C, high-density lipoprotein cholesterol; LDL-C, low-density lipoprotein cholesterol; HbA1c, hemoglobin A1c; hs-CRP, high-sensitivity C-reactive protein. Values are expressed as a mean (SE) or numbers (percentages). The *p*-value was calculated from the ANOVA test for continuous variables and the Chi-square test for categorical variables.

**Table 2 nutrients-14-05298-t002:** Food consumption according to dietary diversity score.

Food Consumption (g/1000 kcal)	Men				Women			
	≤3 (*n* = 524)	4 (*n* = 1156)	5 (*n* = 1144)	*p*-Trend	≤3 (*n* = 297)	4 (*n* = 925)	5 (*n* = 1422)	*p*-Trend
Grains	476.4 ± 2.96	426.3 ± 1.98	384.8 ± 2.00	<0.0001	493.8 ± 4.21	427.6 ± 2.36	378.7 ± 1.92	<0.0001
Whole grains	130.6 ± 6.45	115.2 ± 4.31	108.1 ± 4.37	0.0165	168. 9 ± 9.48	188.1 ± 5.31	166.1 ± 4.32	0.0051
Refined grains	328.2 ± 7.43	289.6 ± 4.97	251.4 ± 5.03	<0.0001	300.7 ± 10.2	211.9 ± 5.75	181.3 ± 4.67	<0.0001
Meat, fish, eggs, and beans	67.4 ± 1.65	79.5 ± 1.10	85.9 ± 1.12	<0.0001	58.3 ± 2.17	72.2 ± 1.21	79.0 ± 0.98	<0.0001
Meat	28.0 ± 0.95	32.0 ± 0.63	33.5 ± 0.64	<0.0001	20.2 ± 1.15	24.0 ± 0.64	26.5 ± 0.52	<0.0001
Fish	17.5 ± 0.69	20.2 ± 0.46	22.8 ± 0.47	<0.0001	14.9 ± 0.94	20.6 ± 0.52	23.5 ± 0.43	<0.0001
Vegetables	175.2 ± 4.38	171.5 ± 2.93	176.7 ± 2.97	0.4517	170.8 ± 5.52	174.3 ± 3.09	175.4 ± 2.51	0.7561
Non-salted vegetables	52.8 ± 2.49	58.2 ± 1.67	71.0 ± 1.69	<0.0001	53.4 ± 3.14	69.3 ± 1.76	81.2 ± 1.43	<0.0001
Salted vegetables	122.4 ± 3.36	113.2 ± 2.25	105.7 ± 2.28	0.0002	117.4 ± 4.20	104.9 ± 2.35	94.2 ± 1.91	<0.0001
Fruits	37.4 ± 3.96	89.3 ± 2.65	143.6 ± 2.69	<0.0001	50.1 ± 7.64	119.2 ± 4.28	185.2 ± 3.48	<0.0001
Milk	5.01 ± 2.19	38.4 ± 1.46	68.6 ± 1.48	<0.0001	6.50 ± 3.79	49.0 ± 2.12	82.2 ± 1.72	<0.0001

Values are expressed as the mean (SE). The *p*-trend was obtained through generalized linear models after adjusting for age, regular exercise, education level, household income, alcohol consumption, smoking status, family history (diabetes, hypertension, and dyslipidemia), menopausal status, and BMI.

**Table 3 nutrients-14-05298-t003:** Nutrient intake according to dietary diversity score.

	Men				Women			
	≤3 (*n* = 524)	4 (*n* = 1156)	5 (*n* = 1144)	*p*-Trend	≤3 (*n* = 297)	4 (*n* = 925)	5 (*n* = 1422)	*p*-Trend
Energy (kcal)	1685.8 ± 19.6	1900.6 ± 15.8	2227.0 ± 15.9	<0.0001	1485.5 ± 33.4	1748.7 ± 18.9	2049.1 ± 15.3	<0.0001
Carbohydrate (g)	181.6 ± 0.71	175.4 ± 0.48	170.8 ± 0.48	<0.0001	189.3 ± 0.93	180.0 ± 0.53	174.3 ± 0.43	<0.0001
Protein (g)	31.5 ± 0.24	33.6 ± 0.16	35.0 ± 0.16	<0.0001	30.1 ± 0.32	33.3 ± 0.18	35.0 ± 0.15	<0.0001
Fat (g)	14.3 ± 0.24	16.7 ± 0.16	18.7 ± 0.16	<0.0001	11.4 ± 0.32	15.0 ± 0.18	17.5 ± 0.15	<0.0001
Calcium (g)	174.4 ± 3.51	221.7 ± 2.37	268.4 ± 2.38	<0.0001	169.9 ± 5.49	238.2 ± 3.11	285.9 ± 2.51	<0.0001
Phosphorus (mg)	469.1 ± 3.38	504.9 ± 2.27	534.2 ± 2.20	<0.0001	466.1 ± 4.89	529.9 ± 2.77	557.1 ± 2.24	<0.0001
Iron (mg)	4.84 ± 0.06	5.22 ± 0.04	5.62 ± 0.04	<0.0001	4.84 ± 0.08	5.56 ± 0.05	5.96 ± 0.04	<0.0001
Potassium (mg)	1089.1 ± 13.90	1210.8 ± 9.36	1337.7 ± 9.41	<0.0001	1082.3 ± 21.06	1286.7 ± 11.93	1422.8 ± 9.62	<0.0001
Vitamin A (µg RE)	227.4 ± 6.81	260.1 ± 4.59	301.4 ± 4.61	<0.0001	205.0 ± 9.41	258.0 ± 5.33	293.1 ± 4.30	<0.0001
Sodium (mg)	1750.8 ± 31.7	1705.7 ± 21.3	1673.6 ± 21.4	0.4080	1670.1 ± 41.1	1642.6 ± 23.3	1585.0 ± 18.8	0.3174
Vitamin B1 (mg)	0.60 ± 0.006	0.63 ± 0.004	0.67 ± 0.004	<0.0001	0.55 ± 0.007	0.62 ± 0.004	0.65 ± 0.003	<0.0001
Vitamin B2 (mg)	0.42 ± 0.005	0.49 ± 0.003	0.56 ± 0.003	<0.0001	0.39 ± 0.008	0.49 ± 0.004	0.57 ± 0.004	<0.0001
Niacin (mg)	7.75 ± 0.07	8.02 ± 0.05	8.24 ± 0.05	<0.0001	7.29 ± 0.09	7.88 ± 0.05	8.22 ± 0.04	<0.0001
Vitamin C (mg)	43.5 ± 1.24	54.1 ± 0.84	68.8 ± 0.84	<0.0001	46.8 ± 2.16	62.4 ±1.23	76.4 ± 0.99	<0.0001
Zinc (µg)	4.21 ± 0.05	4.44 ± 0.03	4.69 ± 0.03	<0.0001	3.98 ± 0.06	4.33 ± 0.04	4.58 ± 0.03	<0.0001
Vitamin B6 (mg)	0.84 ± 0.008	0.89 ± 0.005	0.94 ± 0.005	<0.0001	0.84 ± 0.011	0.92 ± 0.006	0.97 ± 0.005	<0.0001
Folate (µg)	108.4 ±1.85	117.2 ±1.24	128.5 ± 1.25	<0.0001	110.0 ± 2.59	127.0 ± 1.47	136.5 ± 1.18	<0.0001
Retinol (µg)	17.9 ± 0.94	31.3 ±0.63	42.3 ±0.64	<0.0001	16.2 ± 1.42	31.6 ± 0.80	44.1 ± 0.65	<0.0001
β-Carotene (µg)	1224.9 ± 42.7	1336.2 ± 28.8	1530.0 ± 28.9	<0.0001	1099.9 ± 58.1	1324.8 ± 32.9	1462.4 ± 26.6	<0.0001
Fiber (g)	2.32 ± 0.05	3.36 ± 0.03	3.50 ± 0.03	<0.0001	3.40 ± 0.07	3.67 ± 0.04	3.78 ± 0.03	<0.0001
Vitamin E (mg)	4.02 ± 0.06	4.47 ± 0.04	4.96 ± 0.04	<0.0001	3.94 ± 0.09	4.58 ± 0.05	5.18 ± 0.04	<0.0001
Cholesterol (mg)	64.4 ± 2.08	86.4 ± 1.40	100.7 ± 1.41	<0.0001	58.3 ± 2.95	84.9 ± 1.67	101.3 ± 1.35	<0.0001

Values are expressed as the mean (SE). Nutrient intakes were expressed per 1000 kcal. The *p*-trend was obtained through generalized linear models after adjusting for age, regular exercise, education level, household income, alcohol consumption, smoking status, family history (diabetes, hypertension, and dyslipidemia), menopausal status, and BMI.

**Table 4 nutrients-14-05298-t004:** Hazard ratios (HRs) and 95% confidence intervals (CIs) for the risk of metabolic syndrome and its components according to dietary diversity score.

	Men				Women			
	≤3 (*n* = 524)	4 (*n* = 1156)	5 (*n* = 1144)	*p*-Trend	≤3 (*n* = 297)	4 (*n* = 925)	5 (*n* = 1422)	*p*-Trend
Metabolic syndrome, % (*n*)	197 (37.6)	367 (31.8)	361 (31.6)	0.025	113 (38.1)	336 (36.3)	501 (35.2)	0.677
Model 1 ^a^	Ref.	0.82 (0.69–0.98)	0.70 (0.58–0.85)	0.001	Ref.	0.93 (0.75–1.16)	0.81 (0.65–1.00)	0.018
Model 2 ^b^	Ref.	0.83 (0.70–0.99)	0.74 (0.61–0.89)	0.002	Ref.	1.08 (0.87–1.35)	1.03 (0.83–1.29)	0.969
Model 3 ^c^	Ref.	0.83 (0.70–0.99)	0.76 (0.63–0.92)	0.006	Ref.	1.08 (0.87–1.34)	1.04 (0.84–1.29)	0.942
Abdominal obesity, % (*n*)	175 (33.4)	322 (27.9)	322 (28.2)	0.065	189 (63.6)	510 (55.1)	794 (55.8)	0.083
Model 1	Ref.	0.81 (0.67–0.97)	0.69 (0.56–0.84)	0.001	Ref.	0.76 (0.64–0.90)	0.63 (0.53–0.74)	<0.001
Model 2	Ref.	0.84 (0.70–1.02)	0.75 (0.61–0.92)	0.006	Ref.	0.88 (0.74–1.04)	0.80 (0.67–0.95)	0.010
Model 3	Ref.	0.84 (0.70–1.01)	0.76 (0.62–0.93)	0.009	Ref.	0.87 (0.73–1.03)	0.79 (0.67–0.94)	0.007
Elevated blood pressure, % (*n*)	300 (57.3)	604 (52.3)	587 (51.3)	0.039	133 (44.8)	367 (39.7)	545 (38.3)	0.058
Model 1	Ref.	0.88 (0.76–1.01)	0.80 (0.69–0.94)	0.005	Ref.	0.88 (0.72–1.08)	0.80 (0.66–0.98)	0.025
Model 2	Ref.	0.90 (0.78–1.03)	0.87 (0.74–1.01)	0.081	Ref.	1.01 (0.83–1.24)	1.03 (0.84–1.26)	0.804
Model 3	Ref.	0.88 (0.77–1.02)	0.85 (0.73–0.99)	0.052	Ref.	0.98 (0.80–1.20)	0.99 (0.81–1.22)	0.979
Hypertriglyceridemia, % (*n*)	279 (53.2)	444 (38.4)	451 (39.4)	0.035	108 (36.4)	210 (22.7)	316 (22.2)	0.339
Model 1	Ref.	0.86 (0.75–1.00)	0.79 (0.68–0.93)	0.004	Ref.	0.96 (0.77–1.00)	0.88 (0.71–1.10)	0.170
Model 2	Ref.	0.86 (0.74–1.00)	0.79 (0.68–0.92)	0.004	Ref.	1.00 (0.80–1.25)	0.94 (0.75–1.18)	0.441
Model 3	Ref.	0.88 (0.76–1.02)	0.83 (0.71–0.97)	0.023	Ref.	1.00 (0.80–1.25)	0.94 (0.75–1.18)	0.430
Elevated fasting glucose, % (*n*)	222 (42.4)	444 (38.4)	451 (39.4)	0.396	80 (26.9)	210 (22.7)	316 (22.2)	0.140
Model 1	Ref.	0.92 (0.78–1.08)	0.88 (0.74–1.04)	0.139	Ref.	0.86 (0.66–1.11)	0.79 (0.61–1.02)	0.068
Model 2	Ref.	0.93 (0.79–1.09)	0.91 (0.76–1.08)	0.283	Ref.	0.92 (0.70–1.19)	0.87 (0.67–1.14)	0.322
Model 3	Ref.	0.94 (0.79–1.10)	0.94 (0.79–1.12)	0.549	Ref.	0.89 (0.69–1.16)	0.86 (0.66–1.12)	0.286
Reduced HDL cholesterol, % (*n*)	271 (51.7)	584 (50.5)	557 (48.7)	0.220	207 (69.7)	631 (68.2)	977 (68.7)	0.890
Model 1	Ref.	1.06 (0.92–1.23)	0.98 (0.84–1.15)	0.630	Ref.	0.98 (0.84–1.15)	0.95 (0.81–1.11)	0.464
Model 2	Ref.	1.07 (0.93–1.24)	1.01 (0.86–1.18)	0.864	Ref.	1.01 (0.86–1.19)	1.01 (0.86–1.19)	0.935
Model 3	Ref.	1.07 (0.93–1.24)	0.99 (0.84–1.15)	0.651	Ref.	1.03 (0.88–1.21)	1.03 (0.88–1.21)	0.736

^a^ Model 1 was adjusted for age, energy intake, and BMI; ^b^ Model 2 was adjusted for age, energy intake, education level, household income, and BMI; ^c^ Model 3 was adjusted for age, energy intake, regular exercise, education level, household income, alcohol consumption, smoking status, family history (diabetes, hypertension, and dyslipidemia), menopausal status, and BMI; Ref., reference category; HDL, high-density lipoprotein; BMI, body mass intake.

## Data Availability

Data are available from the Korean Genome and Epidemiology Study (KoGES; 4851–032), conducted by the Korea Centers for Disease Control and Prevention (KCDCP), the National Research Institute of Health, and the Ministry for Health and Welfare, Republic of Korea.

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
