# Peer review of "Association between Dietary Diversity Score and Metabolic Syndrome in Korean Adults: A Community-Based Prospective Cohort Study"

_nutrients, 2022, doi:10.3390/nu14245298_

Round 1

Reviewer 1 Report (Previous Reviewer 2)

All the concerns have been fully addressed and it is acceptable for publication in the present form.

Reviewer 2 Report (New Reviewer)

This study confirmed the association between dietary diversity score(DDS) and metabolic syndrome(MetS) in middle-aged adult Korean men and women. The correlation that the higher DDS, the lower possibility of MetS, was confirmed through the cross-sectional analysis, and although it is a study only for Koreans, I believe the same results can be confirmed in regions or countries with different dietary habits. It can be found these limitations are sufficiently mentioned in the Discussion, so it is difficult to point out as a lack of research design. I accept this manuscript to be published as the current version.

This manuscript is a resubmission of an earlier submission. The following is a list of the peer review reports and author responses from that submission.

Round 1

Reviewer 1 Report

The manuscript, “Association Between Dietary Diversity Score and Metabolic Syndrome in Korean Adults: A Community-Based Prospective Cohort Study” by Jiyeon Kim , Minji Kim , Yoonjin Shin , Jung-Hee Cho , Donglim Lee , Yangha Kim examines the gender-stratified association between dietary diversity score (DDS) and risk of metabolic syndrome (MetS)

Critique

1.     The study lasted for 12 years.  Food consumption was only evaluated in the first year of the study.  It is possible that the participants food choices and subsequent changes and weight may have changed during the time of the study and this could have altered the results. The authors admit that this is a major flaw. “…, we assessed food  consumption only at baseline and did not determine whether the dietary patterns of participants had changed throughout the follow-up.”

It may be reasonable to check the weights of the participants during the 12 years of the study to see if weight change over time may be a potential confounding variable.  The subjects did have periodic evaluations during which their weight may have been measured and recorded.

2.      You describe a difference in metabolic syndrome in men but not in women.  “…this prospective study showed that a higher DDS was associated with a lower risk of incident MetS and its components (i.e., abdominal obesity, hypertriglycidemia) in men.” 

This finding was not detected in women.  While you suggest that this gender difference has been reported before, you do NOT suggest a possible etiology.  Please rework your discussion of this finding toward a possible explanation.  Is there a difference in gender  exercise patterns or some other factor that could possibly account for your results? Please state your findings of differences between men and women in the abstract.

3.     Discussion should be significantly shortened and focused on the main findings in the report.

4.     I am confused. Please clarify your statement in the abstract with a seemingly contradictory statement in the conclusions. I suspect the error is in the abstract please correct.

Abstract    “Prospectively, a higher DDS was significantly associated with MetS risk in men (HR: 0.76, 95% CI: 0.63-0.92, p < 0.01).

In conclusion, this prospective study showed that a higher DDS was associated with 304 a lower risk of incident MetS and its components (i.e., abdominal obesity, hypertriglyceridemia) in men.

5.     The discussion should be significantly shortened and focused on the main findings in the study.  As mentioned above the disparate findings between men and women needs to be addressed.

Author Response

Manuscript Number: nutrients-2009087

Title: Association Between Dietary Diversity Score and Metabolic Syndrome in Korean Adults: A Community-Based Prospective Cohort Study

Responses to Reviewer's comments (Reviewer 1)

We thank the reviewer for careful reading and description about our manuscript with the valuable comments. We worked to the best of our abilities to revise the issues reviewer point out.

  1. The study lasted for 12 years. Food consumption was only evaluated in the first year of the study. It is possible that the participants food choices and subsequent changes and weight may have changed during the time of the study and this could have altered the results. The authors admit that this is a major flaw. “…, we assessed food consumption only at baseline and did not determine whether the dietary patterns of participants had changed throughout the follow-up.” It may be reasonable to check the weights of the participants during the 12 years of the study to see if weight change over time may be a potential confounding variable. The subjects did have periodic evaluations during which their weight may have been measured and recorded.

Response: We truly appreciate the reviewer's suggestion. First of all, as mentioned above, food consumption was measured once, at baseline of the study, concerning the individual’s dietary intake over the past year. We indicated changes in weight during follow-up in the Supplementary Table 1 and 2. There were no significant differences in weight change during the 12 years, except for groups with DDS=4. Also, we mentioned contents in the results (line 186-188).

  1. You describe a difference in metabolic syndrome in men but not in women. “…this prospective study showed that a higher DDS was associated with a lower risk of incident MetS and its components (i.e., abdominal obesity, hypertriglycidemia) in men.” This finding was not detected in women. While you suggest that this gender difference has been reported before, you do NOT suggest a possible etiology. Please rework your discussion of this finding toward a possible explanation. Is there a difference in gender exercise patterns or some other factor that could possibly account for your results? Please state your findings of differences between men and women in the abstract.

Response: Thank you for your comments, we suggested a possible etiology that could account for the disparate findings between men and women. We reworked discussion part as follows. Also, we stated differences between men and women in the abstract.

- line 288-294

MetS is characterized by a constellation of cardiac risk factors that include abdominal obesity, atherogenic dyslipidemia, hypertension, and insulin resistance. There are several remarkable features of the MetS in women. Menopause heralds a reduction in circulating levels of estrogen, which possibly affects adiposity, lipid metabolism, and prothrombotic state, consequently contributing to increase cardiovascular risk [53]. These features might relate to differences between men and women in the development of MetS.

  1. Discussion should be significantly shortened and focused on the main findings in the report.

Response: As the reviewer stated, we shortened discussion part and focused on the main findings. We deleted line 268-272 and line 289-295 in the original version of the manuscript.

  1. I am confused. Please clarify your statement in the abstract with a seemingly contradictory statement in the conclusions. I suspect the error is in the abstract please correct. Abstract “Prospectively, a higher DDS was significantly associated with MetS risk in men (HR: 0.76, 95% CI: 0.63-0.92, p < 0.01). In conclusion, this prospective study showed that a higher DDS was associated with a lower risk of incident MetS and its components (i.e., abdominal obesity, hypertriglyceridemia) in men.

Response: There was confusion in our statement. As the reviewer stated, we revised that part.

- line 21-23

Prospectively, a higher DDS was significantly associated with a lower risk of MetS in men (HR: 0.76, 95% CI: 0.63-0.92, p < 0.01).

  1. The discussion should be significantly shortened and focused on the main findings in the study. As mentioned above the disparate findings between men and women needs to be addressed.

Response: As the reviewer stated, we shortened discussion part and focused on the main findings. We deleted line 268-272 and line 289-295 in the original version of the manuscript. Also, we addressed the disparate findings between men and women in the discussion.

Reviewer 2 Report

In the present study, authors investigated the relation between Dietary Diversity Score and Metabolic Syndrome in Korean Adults. Data reveal that higher DDS is associated with lower MS, especially in male adults. The whole study was clearly designed and conducted. The general conclusion is well supported by all data. But some points should be addressed to improve the quality of this study.

1. is the resident location (urban and rural) impact on the data?

2. why adults with younger age were not included? The mean age is around 50.

3. is this study followed for 12 years? There should be certain changes during the period.

4. in this study, it is difficult to discern what is the cause. There are many factors that may impact on the observed result, such as education, income. Why claim that the DDS is the main reason?

5. in fact, it is commonly known that eating in more varied pattern is healthier than only eating limited food category. So what is the significance of this study?

Author Response

Manuscript Number: nutrients-2009087

Title: Association Between Dietary Diversity Score and Metabolic Syndrome in Korean Adults: A Community-Based Prospective Cohort Study

Responses to Reviewer's comments (Reviewer 2)

We thank the reviewer for careful reading and description about our manuscript with the valuable comments. We worked to the best of our abilities to revise the issues reviewer point out.

  1. is the resident location (urban and rural) impact on the data?

Response: According to our previous study: Lee, S., Shin, Y., & Kim, Y. (2018). Risk of metabolic syndrome among middle-aged Koreans from rural and urban areas. Nutrients, 10(7), 859. https://doi.org/10.3390/nu10070859, we agreed that there are differences in the prevalence of MetS depending on the region. However, in this study, we focused on the association between dietary quality and MetS for participants, living in both urban and rural areas.

  1. why adults with younger age were not included? The mean age is around 50.

Response: As we mentioned in the materials and methods, the Ansan-Ansung cohort study, which is an ongoing prospective study to examine the risk and burden of chronic disease in middle-aged and older Korean adults, consists of men and women aged 40-69 years.

  1. is this study followed for 12 years? There should be certain changes during the period.

Response: We indicated changes in metabolic syndrome components, such as waist circumference, blood pressure, triglyceride, fasting glucose, and HDL-C during follow-up in the Supplementary Table 1 and 2. Also, we mentioned contents in the results (line 186-188).

  1. in this study, it is difficult to discern what is the cause. There are many factors that may impact on the observed result, such as education, income. Why claim that the DDS is the main reason?

Response: The multivariable Cox proportional hazard model was used to assess the independent association between DDS and risk of MetS. We adjusted potential confounders, such as age, regular exercise, education level, household income, alcohol consumption, smoking status, family history, menopausal status, and BMI, to evaluate whether DDS is independently associated with MetS risk.

  1. in fact, it is commonly known that eating in more varied pattern is healthier than only eating limited food category. So what is the significance of this study?

Response: This study suggested that a higher DDS was longitudinally associated with a lower risk of MetS and its components. It is the first to investigate the associations between DDS and MetS in Korean adults in a prospective study with long-term follow-up.

Round 2

Reviewer 1 Report

Thank you for including the supplemental data.  The subjects were evaluated every 2 years.  When in the context of this study were  the follow-up subjects' weights obtained?  

Author Response

Manuscript Number: nutrients-2009087

Title: Association Between Dietary Diversity Score and Metabolic Syndrome in Korean Adults: A Community-Based Prospective Cohort Study

Responses to Reviewer's comments (Reviewer 1)

We thank the reviewer for careful reading and description about our manuscript with the valuable comments. We worked to the best of our abilities to revise the issues reviewer point out.

  1. Thank you for including the supplemental data. The subjects were evaluated every 2 years. When in the context of this study were the follow-up subjects' weights obtained?

Response: As we mentioned in the materials and methods, this study was based on the data from the baseline (2001-2002) through the sixth follow-up examination (2013-2014) of the KoGES_Ansan and Ansung study. Therefore, the follow-up subjects’ weights data were obtained from the sixth follow-up examination between 2013 and 2014.

Reviewer 2 Report

All concerns have been addressed.

Author Response

We thank the reviewer for careful reading about our manuscript.